# A User-Centric 3D-Printed Modular Peristaltic Pump for Microfluidic Perfusion Applications

**DOI:** 10.3390/mi14050930

**Published:** 2023-04-25

**Authors:** Jorge A. Cataño, Steven Farthing, Zeus Mascarenhas, Nathaniel Lake, Prasad K. D. V. Yarlagadda, Zhiyong Li, Yi-Chin Toh

**Affiliations:** 1School of Mechanical, Medical and Process Engineering, Queensland University of Technology, Brisbane 4000, Australia; jorgealberto.amayacatano@hdr.qut.edu.au (J.A.C.);; 2Centre for Biomedical Technologies, Queensland University of Technology, Kelvin Grove 4059, Australia; 3School of Engineering, University of Southern Queensland, Springfield Central 4300, Australia; 4Max Planck Queensland Centre (MPQC) for the Materials Science of Extracellular Matrices, Queensland University of Technology, Kelvin Grove 4059, Australia; 5Centre for Microbiome Research, Queensland University of Technology, Woolloongabba 4102, Australia

**Keywords:** 3D printing, modular, peristaltic pump, microfluidic perfusion

## Abstract

Microfluidic organ-on-a-chip (OoC) technology has enabled studies on dynamic physiological conditions as well as being deployed in drug testing applications. A microfluidic pump is an essential component to perform perfusion cell culture in OoC devices. However, it is challenging to have a single pump that can fulfil both the customization function needed to mimic a myriad of physiological flow rates and profiles found in vivo and multiplexing requirements (i.e., low cost, small footprint) for drug testing operations. The advent of 3D printing technology and open-source programmable electronic controllers presents an opportunity to democratize the fabrication of mini-peristaltic pumps suitable for microfluidic applications at a fraction of the cost of commercial microfluidic pumps. However, existing 3D-printed peristaltic pumps have mainly focused on demonstrating the feasibility of using 3D printing to fabricate the structural components of the pump and neglected user experience and customization capability. Here, we present a user-centric programmable 3D-printed mini-peristaltic pump with a compact design and low manufacturing cost (~USD 175) suitable for perfusion OoC culture applications. The pump consists of a user-friendly, wired electronic module that controls the operation of a peristaltic pump module. The peristaltic pump module comprises an air-sealed stepper motor connected to a 3D-printed peristaltic assembly, which can withstand the high-humidity environment of a cell culture incubator. We demonstrated that this pump allows users to either program the electronic module or use different-sized tubing to deliver a wide range of flow rates and flow profiles. The pump also has multiplexing capability as it can accommodate multiple tubing. The performance and user-friendliness of this low-cost, compact pump can be easily deployed for various OoC applications.

## 1. Introduction

Microfluidics devices are increasingly used to establish organ-on-a-chip (OoC) technology [1,2], which has allowed studies on human physiological conditions [3] as well as being deployed in drug testing applications [4]. Medium perfusion in these microfluidic OoC devices must not only ensure efficient nutrient and waste exchange but also mimic physiological vascular flow patterns found in the native environment [5]. Perfusion flow regimes (i.e., recirculating vs. one-pass perfusion) and flow rates have been demonstrated to impact the mass transport of drugs and endogenous soluble molecules to and from target tissues [6]. Fluid-induced shear stress also activates mechano-transduction signaling, which regulates cell signaling and functions [7,8]. Therefore, the fluid pumping system employed in OoC devices should ideally be customizable for perfusion regimes and the range of flow magnitudes and profiles should be extensive enough to recapitulate the diversity of the dynamic physiological environments found in vivo. These include minute interstitial or capillary flows to large circulatory flows in arteries and veins [5,9,10]. From an application perspective, OoC technologies are designed to enable scalable and cost-effective preclinical testing of drug efficacy and toxicity [11]. This implies that the pumping systems used in OoCs must also meet similar utility and upscaling requirements, namely that they have a small footprint, are relatively low cost and simple to operate, as well as being compatible with cell culture operations, such as being sterilizable and able to withstand temperature and humidity in an incubator. 

Currently, there are different pumping systems available for microfluidic perfusion applications. However, options are limited when looking at customizable systems that can deliver a wide range of pumping characteristics while fulfilling the need to be affordable and scalable. Integrated on-chip peristaltic pumps consist of a series of pneumatically actuated Quake-style valves that have been utilized extensively in OoC devices because they are miniaturized and can be readily multiplexed to achieve complex fluid control operations [12,13,14,15,16]. However, these systems are limited to a relatively low flow rate range (6.5 µL min^−1^ to 21.6 µL min^−1^) at high frequency [12] and (0.16 µL min^−1^–2 µL min^−1^) at low frequency [17], which is more comparable to capillary or interstitial flows [18,19,20] and cannot deliver variable flow profiles (e.g., sinusoidal waveform) due to the design of the integrated pumping mechanism. In addition, the need for external air or vacuum supply lines to operate the series of pneumatic diaphragm valves adds manufacturing complexity and increases points of possible leaks and failure [21,22,23]. Gravity-driven perfusion systems are alternatives that can achieve medium perfusion at low flow rates (0.21 µL min^−1^–1.37 µL min^−1^) and for prolonged periods of time. This is achieved by leveraging the potential energy arising from differential liquid heights between inlet and outlet media reservoirs and skillfully designed tilting mechanisms [24,25,26,27]. However, this pumping approach offers operational simplicity at the expense of having precise control overflow characteristics [24,28,29].

External standalone pumps offer the greatest versatility in terms of control over perfusion flow rates and profiles to satisfy the diverse requirements of OoC applications. Commercial peristaltic pumps have been used to obtain control during operation [30], exchangeability [31], flow recirculation [32], and programmability over the flow profiles [33,34] with a wide range of flow rates (e.g., Ismatec pump, 1 µL min^−1^–100,000 µL min^−1^). However, commercially available programmable low-flow peristaltic pumps are expensive [35] and bulky [36], which limits multiplexing operations. The advent of 3D printing technology and open-source programmable electronic controllers [37] presents an opportunity to democratize the fabrication of mini-peristaltic pumps suitable for microfluidic applications at a fraction of the cost of commercial counterparts. Existing 3D-printed peristaltic pumps have mainly focused on demonstrating the feasibility of using 3D printing to fabricate the structural components of the mechanical pump that drives fluid displacement [38,39,40,41]. Thus, they have been mostly developed as an end-product for specific applications with limited end-user customizability [39,42,43]. Ching et al. have developed a DIY peristaltic pump kit (flow rate range from 0.02 µL min^−1^ to 727.3 µL min^−1^), where different 3D-printed modular and standardized components are mounted on an open breadboard to increase the customization capability because different components can be switched out easily [44]. Nonetheless, 3D-printed mini-peristaltic pumps developed to date still lack a fully developed electronic unit with an end user control interface that allows users with no engineering and programming background to easily change pump flow rates and flow profiles. This is because the electronic controllers regulating electrical signals to the pump driver are not often calibrated to a variable (e.g., revolution per minute, RPM of rollers in a peristaltic pump) that is relatable to the volumetric flow rate, thereby necessitating expertise in programming and additional electronic equipment, e.g., a potentiometer to modify pump performance [39] (achieved flow range 700 µL min^−1^ to 6000 µL min^−1^). 

Here, we present a programmable 3D-printed mini-peristaltic pump that provides a similar user experience and customization capability as commercial counterparts, but at a low cost (~USD 175) and with a compact design suitable for perfusion OoC culture applications. The pump consists of a user-friendly, wired electronic module that controls the operation of a peristaltic pump module. This peristaltic pump module comprises an air-sealed stepper motor connected to a 3D-printed peristaltic assembly. The pump design can use different tubing sizes to provide a wide range of operation flow rates and a multi-position tubing housing for multiplexing capability. The electronic control module can also be programmed and pre-set to generate different flow rates and flow profiles by modifying the rotation of the pump assembly. We demonstrate the performance and reliability of a lab-based user-friendly peristaltic pump under different operational conditions to determine its suitability for OoC applications. 

## 2. Materials and Methods

### 2.1. Fabrication of Electronic Control Module

A list of parts for fabricating the electronic control module is listed in Appendix A. The housing for the electronic control module was made by adapting a commercial Acrylonitrile butadiene styrene (ABS) rectangular plastic box (30 mm wide × 68 mm length × 44 mm height) to house the internal circuits and hold the user interface, controls, and wire connections. On the top face of the box, the OLED display, direction switch, and rotary encoder were positioned for easy manipulation by users. A pump 4-pin panel connector, DC power adaptor, and micro-USB port were positioned along the adjacent surfaces of the controller. A female 4-pin panel connector was in the acrylic box that houses the stepper motor and holds the pumping mechanism. The main electronic connections and tools used are shown and described in Appendix A. The power for the microcontroller (Arduino Leonardo, Arduino. cc, Newcastle, Australia) and the motor driver was supplied by the 9 V DC power adapter, while the rotary encoder and OLED display were powered from the microcontroller’s 5 V output pin. A customized PCB (JLCPCB, Shenzhen, China) was designed using electronic design automation software (easyeda-windows-x64-6.5.22. standard edition 2022, EasyEDA, Shenzhen, China) to reduce the number of wire connections and to integrate the microcontroller and a motor driver. Details of the electrical connection and design of the custom PCB are shown in Appendix A. 

### 2.2. Fabrication of Peristaltic Pump Module 

The peristaltic pump module comprised 7 main components: a 3.8 V stepper motor (RS PRO, RS Components Pty Ltd., Bankstown, Australia) that provided the rotational motion, 5 3D-printed parts that secured to the motor shaft and assembled the peristaltic mechanism, and a laser-cut acrylic box that isolated the stepper motor from the outside environment. The components of the pump assembly and motor enclosure were designed using computer-aided design (CAD) software (SOLIDWORKS 2021, Dassault Systems, Vélizy-Villacoublay, France). The CAD models were exported as a Standard Tessellation Language (STL) file and uploaded to ASIGA Composer slicing software (ASIGA Composer Ver. 1.3, ASIGA, Alexandria, Australia) to prepare for printing. The pump assembly parts were 3D printed with a Digital Light Processing (DLP) 3D printer (Asiga Max X27, ASIGA, Alexandria, Australia) with a 385 nm LED light source using a commercial polycarbonate-based resin (Plasclear V2, ASIGA, Alexandria, Australia). Once printed, the parts were soaked and sonicated for 10 min in 98% isopropyl alcohol (IPA, Sigma-Aldrich, Bayswater, Australia), followed by a UV-light post-curing process for 2 h. The pump body, compression rollers, tubing housing, and bearing holder were secured with standard fastening components (M2, M2.5, and M3 screws—Appendix A). The acrylic box was designed to insulate the stepper motor from the moisture and external heat of the incubators. The box was constructed by using a laser cutter (ILS12.75, Universal Laser Systems Inc., Scottsdale, AZ, USA) to cut different parts out of a 3 mm transparent acrylic sheet (Mulford Plastics Pty Ltd., Archerfield, Australia), bonded with solvent cement (Weld-On Adhesives, Inc., Brisbane, Australia), and assembled with the female 4-pin panel connector for the controller wire and connected to the stepper motor.

### 2.3. Programming Interface between Electronic Control and Peristaltic Pump Modules

An open-source microcontroller (Arduino Leonardo, Arduino. cc) was used as the programable interface to send the control signals to the motor driver and provide feedback information on the display to the final user. All algorithms were written in the microcontroller compiler software (Arduino IDE 2.0.3, Arduino. cc). The algorithm to operate the pump at a constant set RPM included the following open-source libraries to enable the communication between the microcontroller and the OLED display: “Wire” library, “Adafruit_GFX” library and “Adafruit_SSD1306” library (the constant RPM program calculations are described in the SI Methods information—A). The programs were compiled and uploaded to the microcontroller through the micro-USB port located on the side of the electronic control box. The motor was controlled by an A3967 Easy Driver stepper motor driver (SparkFun electronics, Boulder, CO, USA). The motor driver operates from 6 to 30 Volts and from 0.15 to 0.7 A with a micro-stepping resolution down to the eighth steps. Given the 200 steps per revolution (1.8° per step) initial motor resolution and the eight micro-stepping driver precision, the final resolution of the motor is 1600 steps per revolution. In addition, the pulsatile flow program also includes the “AccelStepper” library to control the stepper motor’s speed. This library computes the time intervals between steps for the desired acceleration profile and sends the signal to the motor driver (the pulsatile flow program calculations are described in Supplementary Information Methods—B). The connections to the motor driver, rotary encoder and OLED display screen, remained the same as the constant RPM program. 

### 2.4. Validation of Programmed Stepper Motor Rotation 

A rigid pointer was attached to the stepper motor rotor and located close enough to a thin wall plastic box to produce an audible sound with every rotation. The sound was recorded using a microphone and open-source audio processing software (Audacity 3.2.3, The Audacity Team, Limassol, Cyprus). The audio files were recorded to be between 3 and 6 min of continuous audible collisions for each *RPM* rotation value evaluated. The audio file was analyzed to measure the interval between successive collisions and thus determine the revolutions that would occur in a minute of this collision rate. The peaks of the waveform were located, analyzed, and plotted using a MATLAB function (MATLAB, MathWorks, Natick, MA, USA) and the values were stored in an array. The period among all peaks present in the waveform was calculated and then converted to RPM from the following relation:(1)$$RPM_{i}=\frac{60}{T_{i}},$$
where $T_{i}$ is the period between peaks for a set of two consecutive peaks. The average RPM was then calculated along with the standard deviation, and the results of testing expected RPMs in the range of 1 RPM to 30 RPM were plotted. 

To validate the programmability of a pulsatile flow, a pressure sensor (HSCDRRD002NDSA3, Honeywell, Charlotte, NC, USA) was installed in the pump outline tubing. The pressure sensor was connected to a data acquisition system (DAQ) (USB-6009, National Instruments, Sydney, Australia) and the digital output signal was plotted and logged using LabVIEW (LabVIEW 2021, National Instruments, Australia). Sequential cycles of the pressure pulses were recorded and compared to the programmed input to verify concordance in acceleration and deceleration flow rates. Statistical analysis and plotting were made on Prism (version 9.4.1, GraphPad). 

### 2.5. Characterization of Pump Flow Rates and Multiplexing Operation

To characterize the volumetric flow rates delivered by the pump at different RPM settings and different tube sizes, dyed deionized water was perfused into a reservoir on a digital balance with a milligram precision reading (BCE224I-1S, Sartorius Lab instruments Gmbh & Co., Göttingen, Germany) over a fixed time. The measurement of the balance was recorded sequentially over time: 10 min at low RPMs (1–3) and 3 min at higher RPMs (5, 10, 20). The average volumetric flow rate delivered by the pump was calculated by dividing the volume of water displaced during the elapsed time. Pump flow rates were determined for 4 sizes of elastic tubing: 1.3 mm (OD) × 0.5 mm (ID), 1.8 mm (OD) × 1.0 mm (ID), 2.3 mm (OD) × 1.5 mm (ID), 2.8 mm (OD) × 2.0 mm (ID) tubing. The tubing used for testing is a medical-grade soft silicone with durometer 55 Shore A, compatible with peristaltic pumps (Catalogue No. 310-0504, -0118, -1523, -0228, Gecko Optical Company, Perth, Australia). 

The multiplexing operation of the pump was evaluated using the same 4 tubing sizes but only at 1, 5, and 10 RPMs. Three equal-sized tubings were installed in the pump mechanism and connected to independent reservoirs. The mass of water dispensed over time was recorded independently at each tube position. The average volumetric flow of each position was averaged and compared among all positions. All data were collected, analyzed, and plotted using Prism. 

### 2.6. Fabrication of 3D-Printed Microfluidic Vasculature Channel Device

A microfluidic vasculature model was produced using Lumen X bioprinter (CELLINK, Gothenburg, Sweden) and GelMA PhotoInk™ (CELLINK, Gothenburg, Sweden) crosslinked at a wavelength of 401 nm. The design of the vascular channel with bifurcation was similar to our previous work [45] and was bioprinted with a resolution of 100 µm layer thickness and 20 mW-cm^−1^ of projector light power. The 3D-printed GelMA block (16 mm × 7 mm × 3 mm) encasing the vascular channel was stored in phosphate-buffered saline (PBS) at 4 °C until cell seeding. To contain the GelMA block, a 3D-printed bioreactor was designed with computer-aided design (CAD) software (SOLIDWORKS 2021, Dassault Systems), which was then uploaded to ASIGA Composer slicing software (ASIGA Composer Ver. 1.3, ASIGA) for printing. The bioreactor was fabricated with a Digital Light Processing (DLP) 3D printer (Asiga Max X27, ASIGA, Australia), utilizing a commercial polycarbonate-based resin (Plasclear V2, ASIGA, Australia) with a 385 nm LED light source. The housing underwent a 10 min soak and sonication process in 98% isopropyl alcohol (IPA, Sigma-Aldrich, Australia) followed by a UV-light post-curing process for 2 h. To sterilize the device, an 80% ethanol soaking process was utilized for 2 h before the microfluidic circuit assembly.

### 2.7. Cell Culture

Human aortic endothelial cells (HAECs) were cultured in Vascular Cell Basal Medium supplemented with Endothelial Cell Growth Kit-VEGF and 1% penicillin-streptomycin in T175 cell culture flasks at 37.5 °C and 5% CO_2_. For cell seeding in the GelMA microfluidic channel, a cell density of 1 million cells/mL was used. After seeding, the cells were allowed to attach for 12 h before initiating perfusion at a rate of 34 µL/min for 2 weeks.

### 2.8. DAPI and F-Actin Staining

The GelMA microfluidic channels were removed from the 3D-printed bioreactor, and cells were fixed by soaking them with 4% paraformaldehyde (PFA) inside a 2 mL Eppendorf tube followed by incubation at 37 °C for 15 min. Following fixation, cells were permeabilized with 0.5% Triton-X 100 in PBS for 5 min at 37 °C. Staining was performed using 1 mg/mL DAPI stock solution diluted in PBS at a 1:500 ratio and 66 µM F-Actin stock solution diluted at 1:400. The channels were incubated at room temperature for 30 min, followed by three washes with PBS. The HAECs were imaged using a Nikon confocal microscope with a 10× objective.

### 2.9. Statistical Analysis

A minimum of three experimental replicates (*n* ≥ 3, unless otherwise mentioned) were used in each study and the results are presented as mean value ± standard deviation. Data obtained from the multiplexing evaluation were analyzed using one-way ANOVA (GraphPad Prism 9 software, San Diego, CA, USA). Differences between the groups were analyzed using the Tukey test of multiple comparisons and a confidence interval of 95% (*p* < 0.05) was considered statistically significant unless otherwise specified.

## 3. Results

### 3.1. Overview of the Modular and Customizable Mini Peristaltic Pump

The 3D-printed mini-peristaltic pump was designed to be modular to cater to the needs of operating microfluidic perfusion devices under high humidity environments, such as in a cell culture incubator. It consisted of a programmable electronic control module that was connected via a wire to a mechanical pump module (Figure 1). The modular design separated the electronic control components from the mechanical assembly of the pump responsible for physical fluid displacement. The operation and control of the pump can be performed from a remote location away from the mechanical pumping assembly, allowing for the pump module to sit inside the incubator close to the microfluidic device (Figure 1A). The electronic control module was a compact packaged box comprising of external dials and an OLED screen to change and display the variable parameter (i.e., revolutions per minute, RPM) that was calibrated to different pump flow rates as well as external connection ports for power supply, programming, and the pump module. 

The pump module was built as a low-cost and 3D-printed mechanical assembly with a small footprint measuring 114 mm × 55 mm × 55 mm. It consisted of a stepper motor housed within an air-tight acrylic box, which was coupled to an assembly of 3D-printed polycarbonate parts to generate the fluid-driving mechanism. Up to three tubes of varying internal diameters can be readily mounted to the pump assembly to increase the throughput and range of the pump. The modular design combined with the use of water-resistant connections and a motor insulation box provided good protection against humidity, thereby allowing the pump module to be housed inside a cell culture incubator without risk of corrosion and electronic damage. In addition, the compact size of the pump allows it to be easily placed on an incubator shelf unlike bulkier commercial pumps (Figure 1B).

### 3.2. Construction of the Electronic Control Module

The key design objective of the electronic control module was to facilitate end-user manipulation of the pump by packaging and integrating external user-interface components with the electronic control components (i.e., microcontroller and motor driver) within a compact box (Figure 2A). This overcomes the current limitation of existing open-source 3D-printed microfluidic pumps, which rely on direct reprogramming of microcontrollers [41,43,44], as well as additional hardware, such as potentiometers [39,46] and electromagnetism from an external rotating rotor [47,48,49], to change the pump rotation settings. 

Users interact with the electronic control module via the external user interface components, namely a rotary encoder, a direction switch, and an OLED screen. The rotary encoder is a dial that was used to register changes in rotation speed set by the user. The OLED screen was used to display information about the selected motor rotatory speed, and a direction switch gave the option to reverse the rotational direction of the motor shaft (Figure 2A). Three external connections were included, namely a DC power adapter, a micro-USB connection to load any programs to the microcontroller, and a pump four-pin panel connector to join to the mechanical pump module (Figure 2A). 

A customized PCB was designed as an interface between the different external components and the Arduino microcontroller to reduce the wiring and connection density so that all user interface connections and control elements can fit inside a small footprint box not exceeding 130 × 68 × 44 mm. The PCB was customized to have similar dimensions as the microcontroller so it could fit and connect to all the microcontroller pins using pin header connectors, allowing the connections to be made to any input/output pins of the microcontroller via traces on the PCB (Figure 2B). The wires from the rotary encoder, direction switch button, and pump four-pin panel connector were connected through the PCB to the digital pins of the microcontroller while the OLED screen was connected to the PCB using the serial clock and data pins of the microcontroller. The motor driver was soldered directly onto the PCB, while the direction switch, rotary encoder, OLED screen, and other external connections used ribbon cables with locking headers to allow the control box external components to be easily removed and reattached if disassembly or repairment of the controller module was required (Figure 2B). 

### 3.3. Programming and Validating the Electronic Control Module

The electronic control module was designed to provide users with the freedom to program the pump for specific applications while having direct manual control of pump flow rates during everyday operation. We first characterized the function of the electronic control module using a constant flow rate program (Figure 3A–C). This program enabled users to manually input a specific flow rate output by selecting an RPM value using the rotary encoder. The program would read the user input from the rotary encoder and direction switch to determine the rotation speed and direction, respectively, and then respond by changing the speed of the stepper motor rotation and displaying the RPM value on the OLED screen (Figure 3A). The rotation speed of the stepper motor depended on the rate at which the motor performed its steps (here we used a motor with 200 steps per revolution). The program calculated the time interval between each step to produce the RPM setting selected with the rotary encoder, which could adjust the speed in 1 RPM increments. The program then instructed the motor driver to deliver electrical current through the coils of the motor at a calculated step rate; additionally, the motor driver managed the motor direction changes through communication between the microcontroller whenever the direction switch was pressed. When the controller was first powered on, the software initialized all variables to predefined values (i.e., counterclockwise rotation at 1 RPM). To validate the accuracy of the calculated motor step time intervals calculated by the algorithm, the rotation frequency of the motor shaft was measured at different RPM settings set by the rotary encoder. The time taken to complete one revolution was determined using sound recording software that logged an audible sound of an element attached to the motor shaft hitting a static objective located in the same position for every revolution (Figure 3B). We evidenced a precise correlation between the set RPM and measured RPM values (R^2^ = 0.9998) (Figure 3C), indicating that the electrical setup worked accordingly to the calculated motor step time intervals.

Pulsatile and oscillatory flows have emerged as a solution to overcome limitations in microfluidic applications, where steady flows are not sufficient for specific conditions and applications, often pushing the common materials for microfluidic setups to their limits [50]. These types of flows differ from steady flows due to their time variation in terms of the pressure gradient, resulting in a wavering flow. Delta pressure can be used to describe these types of flows in terms of the amplitude, frequency, and time of the pressure oscillation. In biological experimentation, these flow profiles are of significant relevance because most biological processes in macroscopic living organisms are based on pulsatile fluid transport. Notably, the growth and viability of mechanosensory cells have been improved using pulsatile and oscillatory flows [10,51,52]. An important application in biology is simulating heartbeat pulses to drive fluid in cell culture applications for cardiovascular research [53,54]. 

To demonstrate the programmability function of the electronic control module, a pulsatile flow program was developed (Figure 3D–F). The pulsatile flow was defined by a maximum RPM value and a desired beats per minute (BPM) selection, which were chosen as the two main input values that had been programmed for user interface manipulation in the software (Figure 3D). By choosing these values, calculations can be made to determine the “AccelRate” (Acceleration rate) and “AccelTime” (time between acceleration rate changes) variables from the “AccelStepper” library to obtain the acceleration ramps used in the algorithm. A set of calculations were developed for both the acceleration phase of the pulse and the deceleration phase of the pulse. For these calculations, it was assumed that the desired output wave takes the form of a sawtooth wave, and thus the decrease segment is significantly shorter than the increase (Figure 3E). 

To characterize the corresponding pulsatility of the flow that was delivered by the pump, a pressure sensor located in the line of the pump discharge was used to capture the pressure changes generated. The pulsation was recorded several times, plotted, and overlapped to verify the consistency of the timing and pressure. We used an acceleration time of 250 ms, which produced an acceleration rate equal to 25 steps per second (AccelTime = 250 ms, AccelRate = 25). The results of the experiments showed a maximum amplitude of approximately 1 mm H_2_O with a two-second pulse width and a period of approximately 15 s which coincided with the programmed pulse delay in the algorithm (Figure 3F). A standard deviation of ~0.15 mm H_2_O of pressure was found among the experimental evaluations; however, the period and amplitude of the pulses were consistent with the programmed ramp. 

### 3.4. Construction of the Mechanical Pump Module

The pump module was designed to be a miniaturized roller-type peristaltic pump that can work inside a high-humidity environment such as an incubator. It comprised a laser-cut acrylic box that isolated the stepper motor from moisture. The rest of the pump parts shape the mechanical assembly of the peristaltic pump (Figure 4A,B). A motor shaft adaptor (part #3) was assembled on the stepper motor shaft to hold three compression rollers (part #6) and transmit the rotational movement of the motor. The three compression rollers were distributed 120 degrees from each other. The pump body (part #2) was secured to the acrylic box, providing support to the other end of the motor shaft adapter and the necessary space to secure the tube housing (part #5). The tube housing functioned as a stator to compress the tubing against the rotation of each compression cylinder and also to delineate the tube positions in conjunction with the pump body. The end of the pump body was connected to a bearing holder (part #4) to support the motor shaft adapter sub-assembly (Figure 4A). A 6 mm inner diameter bearing (Appendix A) was installed in the bearing holder to stabilize the entire length of the motor shaft sub-assembly. This is to ensure that tubing at various positions along the motor shaft is subjected to similar compression forces and, thus, generates similar flow rates. The repeated compression and relaxation of the elastic tubing against the tube housing provided a positive displacement force to push the fluid in the direction of the rotational movement (Figure 4C). Except for the stepper motor and the acrylic housing, all parts were fabricated by SLA 3D printing using a polycarbonate-like resin (Figure 4D). The fidelity of each 3D-printed component was evaluated to ensure the peristaltic mechanical assembly would be functional. To this end, the critical dimensions of each printed part were measured and compared to that of the designed CAD model (Appendix A). The average printing errors of the measured dimensions of each part were quantified and plotted. It was observed that there were no significant differences between the printed parts and the designed CAD models, maintaining an average printing error below 1.5% (Figure 4E). This indicated that the 3D-printed parts could be produced with good fidelity to guarantee the assembly and functionality of the mechanism of the pump. 

### 3.5. Characterization of Pump Performance and Flow Rate Range

To evaluate the pump performance, we first characterized the fluid delivery with different tube sizes and different rotation speed (i.e., RPM) settings using the constant flow rate program. The choice of tubing size used in the mechanical pump module and the RPM setting in the control module will collectively determine the volumetric flow rate that the pump can deliver. Four tubing sizes ranging from 0.5–2 mm internal diameter (ID) were evaluated. The results of the characterization showed a consistent flow rate over the sequential increase in RPMs. We observed a linear increase in flow rates with increasing RPM settings (1–20 RPM) for different tubing sizes (Figure 5A). An equation relating RPM settings to the measured flow rate by linear regression for each tube size was obtained to help users select the RPM setting that would be needed to achieve a desired flow rate (Figure 5B). Here, the stepper motor selected had a small step size (i.e., 1.8° per step) to increase its precision when rotating. However, the torque of the motor is reduced when ramping the rotational speed. Therefore, the maximum rotation speed evaluated was 20 RPM, which produced flow rates of 1.75 mL/min, 1.0 mL/min, 0.55 mL/min, and 0.09 mL/min with 2 mm, 1.5 mm, 1.0 mm, and 0.5 mm ID tubing, respectively (Figure 5A). Nonetheless, we showed that by using four tubing sizes ranging from 0.5–2 mm ID and changing the rotational speed setting between 1–20 RPM, the 3D-printed mini-peristaltic pump could reliably deliver a relatively wide range of flow rates from 0.00358 mL/min to 1.75 mL min^−1^. The characteristic flow curves of different tubes facilitate user selection of the tubing and RPM setting according to the adequate volumetric flow for a specific application. 

Next, we evaluated the multiplexing capability of the mini-peristaltic pump to support running parallel microfluidic devices. The tube housing and pump body were designed with the necessary space to house three parallel tubes along the motor shaft adapter. We evaluated the precision and consistency of flow delivery among the three positions simultaneously by pumping through a set of three tubes with the same ID (0.5 mm, 1.0 mm, 1.5 mm, and 2.0 mm) at three different rotational speed settings (i.e., 1, 5, and 10 RPM) (Figure 5C). Two-way ANOVA statistical analysis showed small statistically significant differences among the flow rates delivered by some tubes across the different speed settings (Figure 5C). Despite the minor differences in between some tubing positions, results of the test showed consistent and synchronized flow perfusion below 5% difference in volumetric flow delivered at any speed or tubing used. The maximum volumetric standard deviation was 18.6 µL/min for the 2 mm ID tubing at 10 RPM, which corresponded to a relative error of 2.1%. The result showed a reliable pumping mechanism with a maximum evidenced error of 4.2% at 10 RPM with 1 mm ID tubing (Table 1).

To evaluate the performance of a 3D-printed pump under realistic conditions of high temperature and humidity in an incubator, a microfluidic perfusion culture system was designed and implemented (Figure 6A). The microfluidic cell culture device was an adaptation of a 3D bioprinted GelMA-based vascular channel, which was previously developed by our group [45]. The 3D-printed vascular channel was fitted inside a 3D-printed bioreactor to facilitate connectivity (Figure 6B) to the peristaltic pump and medium reservoir in a perfusion circuit. Human aortic endothelial cells (HAECs) were seeded in the microfluidic vascular channel and were continuously perfused at a flow rate of 34 µL/min using the 1 RPM rotation setting with 1 mm ID tubing on the peristaltic pump. The pump can be operated continuously for two weeks inside the incubator, during which the operating temperature was consistently maintained within the manufacturer’s recommended range (Appendix A). After 2 weeks of perfusion culture, the HAECs were fixed and stained with DAPI and Phalloidin to visualize the cell coverage of the channel lumen (Figure 6C). The observed coverage and attachment of cells inside the channel, along with nuclei orientation along the flow direction, indicated that the pump was able to drive medium perfusion over the culture period. In conclusion, the experiment’s results validate the pump’s successful operation for cell culture experimentation, indicating its potential use in organs-on-chip applications.

## 4. Discussion

A large variety of fluid-driving pumps for OoC devices has been developed in the last few decades, aiming to provide an accurate and physiologically relevant design specifically for microfluidic applications. In particular, peristaltic pumps have shown good capabilities for flow perfusion while achieving pulsatile waveforms similar to in vivo conditions [55]. In this work, we presented a low-cost mini-peristaltic pump made from standard components, open-source components, and 3D-printed parts, which can also fulfill user-centric requirements, such as changing pump flow rates, programming flow profiles, and performing multiplex perfusion. The final product was a compact design with an electronic control module measuring 130 × 68 × 44 mm and a pump module measuring 115 × 55 × 55 mm, with an estimated cost of around ~USD 175 to manufacture. The presented pump had three key differentiating features that made it ideal for microfluidic OoC applications. First, it is small, low-cost, and has multiplexing capabilities, which differentiates it from the expensive commercial pumps with bulky sizes and from some in-hose pump systems that are only designed for one application or one single perfusion loop [41,43,46,56,57,58]. Second, it can withstand a high-humidity environment due to its modular design, which separates the control module from the pump module and allows for the pump to be controlled from outside the incubator. These two attributes made the pump ideal for multiplexed drug testing operations, as multiple pumps can be concurrently deployed inside a single incubator to run many devices in parallel. Unlike existing 3D-printed peristaltic pumps which rely on the use of microcontrollers, potentiometers, or other control systems to alter pumping characteristics [39,41,43,44,47], the presented 3D-printed mini-peristaltic pump has a user-friendly interface that allows end-users without programming experience to directly change pumping characteristics, such as flow rate and flow direction. The pump also has a programmable function to upload algorithms for a more sophisticated pumping profile. Overall, the presented mini-peristaltic pump was able to fulfil both the cost-efficiency and functional requirements of a standalone microfluidic pump, which has not been achieved in current 3D-printed pumps developed to date or commercial solutions.

The selection of pump components is critical to the pumping performance in terms of flow rate range and flow profiles that can be achieved. The stepper motor is a key component of the peristaltic pump module and therefore directly influences the pump performance. In addition, the power and control components, such as the motor driver, will vary depending on the type and characteristics of the selected motor. High precision rotation is the major feature of stepper motors as compared to other types of motors. However, increasing the rotational speed of a stepper motor has an immediate effect on the torque delivered, meaning that increasing speed will decrease the capacity of moving the same fluid mass until the motor stalls against the fluid load. For this reason, defining a maximum speed achievable for the specific motor selected is essential to guarantee consistent flow perfusion over time from the pump. In addition, the processing time of signals to the motor driver must be considered when increasing the RPMs of the motor, since the speed of the microprocessor running the software loop versus the revolution speed of the motor can affect the precision of the calculated rotation time.

The design and functionality of a roller-type peristaltic pumping mechanism rely on the precision and accuracy of the assembly parts to guarantee the parts are within the dimensional tolerances necessary for the mechanical assembly to work as desired. Thus, the resolution of the 3D printing fabrication process will affect the performance of the pump mechanism. The 3D printer was a Digital Light Processing (DLP) printer and could print with an accuracy of 27 μm pixel resolution, guaranteeing suitable parts for a 0.4 mm wall thickness tubing. The compression components of the peristaltic assembly (i.e., tube housing and compression roller) were designed for 0.4 mm wall thickness elastic tubing. However, the assembly CAD files were designed to allow gap adjustment for different tube dimensions by modifying the outer diameter of the compression cylinder. Finally, a consistent volumetric flow was evidenced on the four different tubes tested; nevertheless, minor variations were shown in some tests due to the dragging of the tube by the rollers on the peristaltic mechanism. Nonetheless, this issue was solved by using tube locks before the mechanism, which is a widely used solution on commercial peristaltic pumps.

## 5. Conclusions

A low-cost mini-peristaltic pump with a user-friendly control interface made from standard components, open-source components, and 3D-printed parts was presented. The user-friendly control interface, which was constructed by packaging and interfacing an open-source microcontroller, motor driver, and off-the-shelf electronic components, allowed users to directly tune flow rates and input pre-programmed flow profiles. Using SLA 3D printing, we could fabricate various components of the peristaltic pump assembly with good accuracy of <5% error from the designed dimensions. The pump showed consistent flow delivery across all tested tubing dimensions and RPMs. We expect our pump to encourage the use of in-house perfusion systems as an alternative for OoC experimental process development. In all, the presented pump can fulfil both the cost-efficiency and functional requirements of a standalone microfluidic pump, which has not been achieved in current 3D-printed pumps developed to date or commercial solutions.

## Figures and Tables

**Figure 1 micromachines-14-00930-f001:**
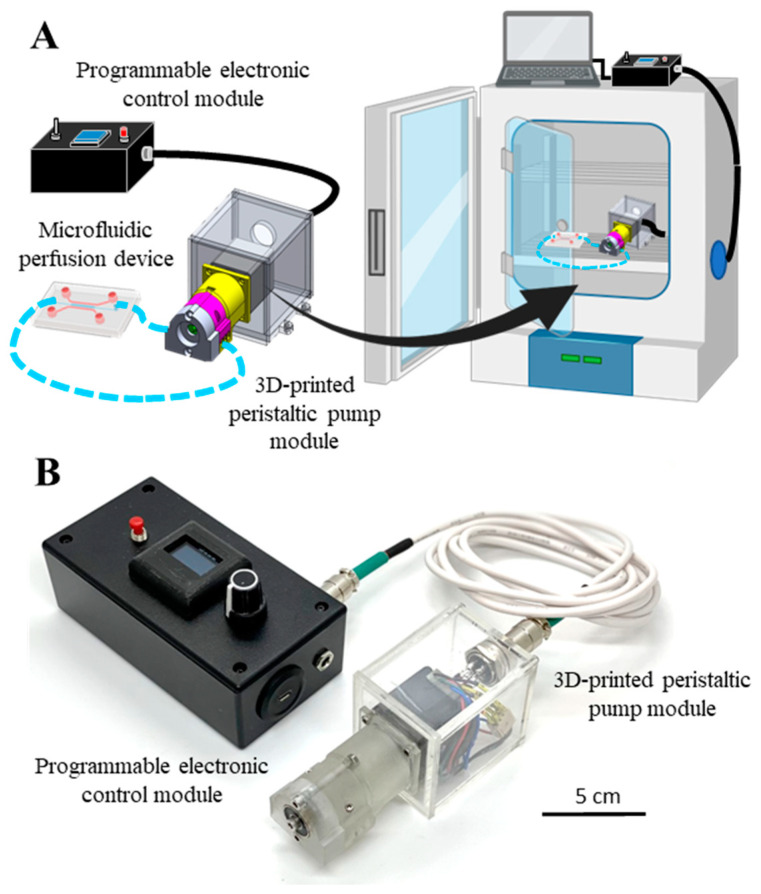
Design of a modular mini-peristaltic pump. (**A**) Concept illustration showing the two modules of the pump—a programmable electronic control module and a 3D-printed peristaltic pump module (left) and their respective placements inside an incubator during microfluidic perfusion culture (right). (**B**) A physical prototype of the modular mini-peristaltic pump.

**Figure 2 micromachines-14-00930-f002:**
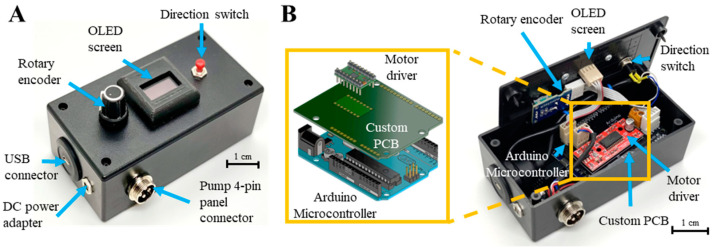
Design and construction of the electronic control module. (**A**) Image of the control box housing external user connections for power input, programming, and motor control. Additionally, the user interface components include a rotary encoder, OLED display, and direction switch. (**B**) Image of the electrical connections, arrangement and wiring of the different components inside the control module box. Inset shows a magnified view illustrating the assembly of the custom PCB, Arduino microcontroller, and motor driver.

**Figure 3 micromachines-14-00930-f003:**
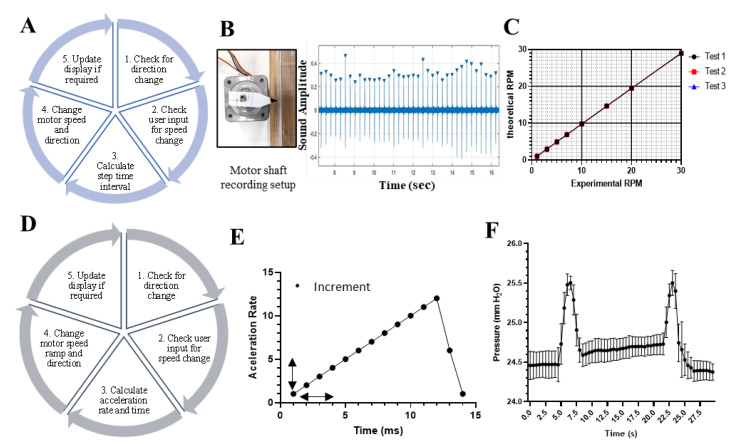
Characterization of the programmable electronic control module of the pump. (**A**) Logic sequence of the algorithm programmed in the microcontroller (Arduino) to deliver a constant flow rate. (**B**) Image of the sound collision setup and plot obtained after processing the sound file recording at an RPM set. (**C**) RPM validation plot of different RPM tests from 1 to 30 versus the theoretical number calculated for the microcontroller (Arduino) software. (**D**) Logic sequence of the algorithm programmed in the microcontroller (Arduino) to produce a pulsatile flow profile. (**E**) Sawtooth wave ramp of acceleration rate and acceleration time programmed in the software to achieve a waveform-type flow profile with the stepper motor. (**F**) Waveform obtained with an acceleration time of 250 ms and an acceleration rate equal to 25 steps per second (AccelTime = 250 ms, AccelRate = 25).

**Figure 4 micromachines-14-00930-f004:**
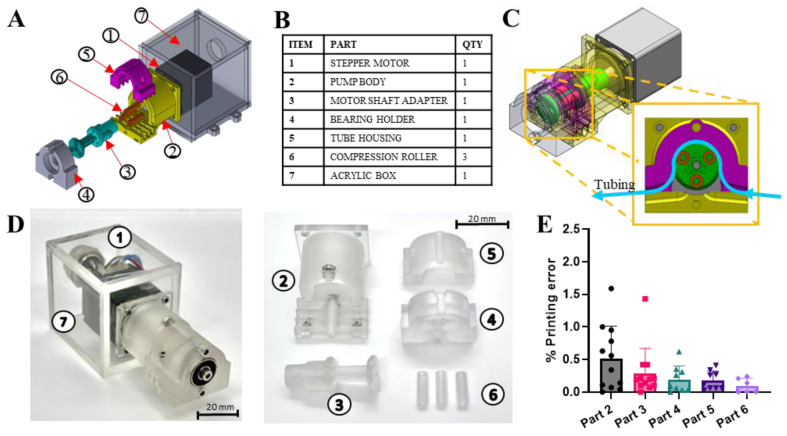
Design and fabrication of the mechanical pump module. (**A**) Exploded view of the pump peristaltic mechanism, motor, and acrylic housing. (**B**) List of main parts and components of the peristaltic mechanism module. (**C**) Pump mechanism assembled view and sectional view of shaft adapter, compression cylinders, and tube housing. (**D**) 3D-printed parts and components (left panel) and full peristaltic assembly (right panel) with motor and acrylic box. (**E**) Accuracy of 3D-printed pump parts and components compared to the original CAD design depicted as a percentage of printing error. The part numbers are related to those described in (**D**), and the number of dimensions evaluated was determined by the geometrical complexity of the part. For instance, the length of the shaft connection and the tube guide, the ID of the shaft housing, and the depth of the tube fixation point were considered for the pump body (Part 2), whereas the compression cylinder (Part 6) was evaluated by its length and ID. Data presented as mean ± SD, *n* = 3.

**Figure 5 micromachines-14-00930-f005:**
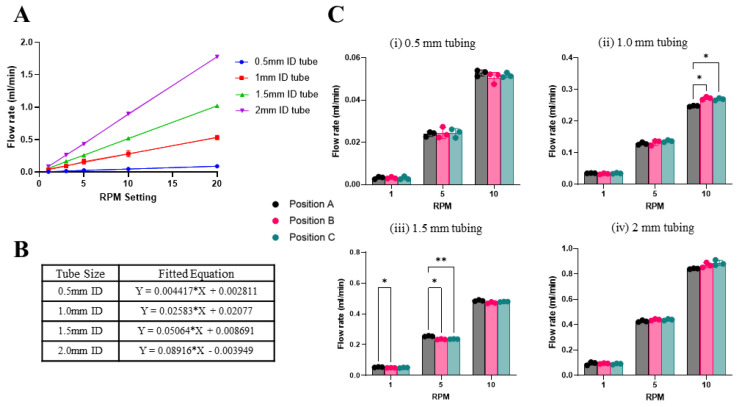
Characterization of pump flow rate and multiplexing capability. (**A**) Summary plot of independent flow rate characterization for four different tubes at different RPM sets. Experiments comprised ten technical replicates of five experimental replicates. Statistical repeats for each RPM tested. (**B**) Linear regression equations relating RPM settings (X) and desired flow rates (Y) for tubing with different internal diameters (ID). (**C**) Plots of the multiplexing results of each tube characterized at 1, 5, and 10 RPM settings. Data presented as mean ± SD, *n* = 3 (** *p* < 0.01; * *p* < 0.05).

**Figure 6 micromachines-14-00930-f006:**
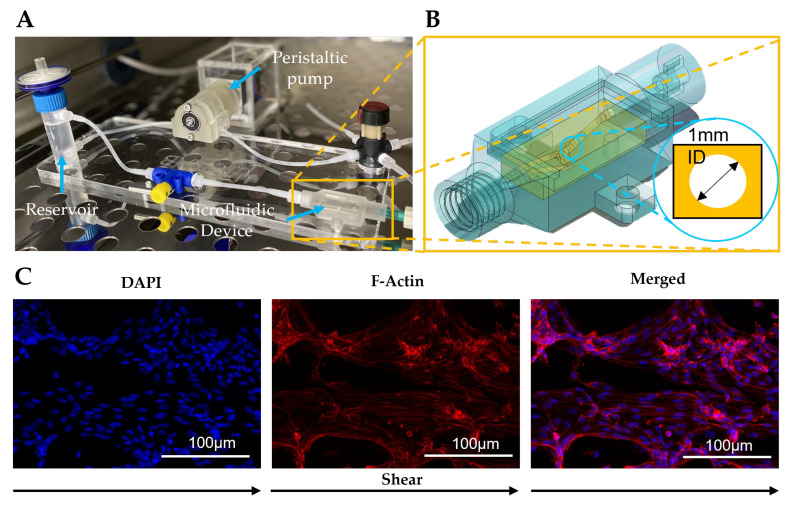
Experimental application of the peristaltic pump inside an incubator while driving media into a microfluidic loop circuit. (**A**) Image of the perfusion circuit assembly with a 3D-printed microfluidic device. (**B**) CAD image of the 3D-printed bioreactor and the microfluidic channel assembly, describing the shape and the inner diameter (1 mm ID) of the channel used for the experiment. (**C**) Fluorescence confocal images of endothelial cells within a microfluidic channel after two weeks of continuous perfusion at 34 µL/min (1 RPM—1 mm tubing ID).

**Table 1 micromachines-14-00930-t001:** Percentage errors relative to the average flow rates delivered by three tubings placed at different locations for specific tubing size and RPM settings.

% Error Relative to the Average Flow Rate of Three Tubings				
RPM	Tubing ID (mm)			
	0.5	1	1.5	2
1	2.6	2.4	3.7	1.8
5	0.9	2.5	3.8	1.3
10	1.8	4.2	1.2	2.1

## Data Availability

The data are available as per request.

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
