# Peer review of "A User-Centric 3D-Printed Modular Peristaltic Pump for Microfluidic Perfusion Applications"

_micromachines, 2023, doi:10.3390/mi14050930_

Round 1
Reviewer 1 Report
Toh et al. reported a 3D-printed peristaltic pump for microfluidics and organ-on-a-chip applications. The pump consists of a wired electronic module for customization, and the air-sealed stepper motor allows for use in high-humidity environments such as a cell culture incubator. The results showed that the pump is able to achieve a wide range of flow rates and profiles. The manuscript is well-written and organized. However, there are some issues that need to be addressed before considering it for publication in Micromachines.
- The authors claimed that the pump is able to work in an incubator, but I do not see any results. They should test the pump in a real incubator and measure the flow profiles.
- Some tests were done in microfluidic devices, but the authors did not provide specific details about the design of the tested device, such as the channel size, or the pressure after connection to the device. This information should be included to provide context for the experiments.
- There are some minor typos, such as on page 2, lines 76-80, where some cited references have a space before the word, while others do not. The authors should thoroughly proofread the manuscript to correct these errors.
- The styles of references should be kept consistent throughout the manuscript. For instance, reference 21 is written as "Lab on a Chip", while reference 23 is written as "Lab Chip". The authors should carefully check the manuscript to ensure consistency in reference formatting.
Overall, the manuscript is well-written and presents an interesting contribution to the field of microfluidics and organ-on-a-chip technology. However, addressing the issues outlined above will further strengthen the quality and clarity of the manuscript.
Author Response
Please refer to reviewers' response document.

Reviewer 2 Report
The manuscript presents the implementation and evaluation of a simple home-made peristaltic pump that can be used in microfluidic applications. The system is of rather small footprint and made up of available electronic and mechanical components. However, there are similar works as mentioned in the manuscripts with no significant differences.
The manuscript is clear and almost well-presented; however, it could be improved in terms of technical information. From the engineering point of view, more technical details are required to be added in the introduction about the other alternatives. For example, the minimum and maximum flow rates provided by other pumps and more specifically what is needed typically in microfluidic systems.
The authors claimed that the current system is resistant to humidity. But there are some concerns in this regard. The stepper motor is contained in an acrylic container. But what is the sealing approach at the shaft output or where the ball-bearing is fixed? The stepper motors generate heat especially when they are used at low RPMs. If the motor is perfectly sealed in an acrylic container, how the generated heat is transferred to the environment? In some cell culture or organ-on-a-chip application the culture may be extended to a month or more. In that case, the generated heat may affect the motor performance.
The cable used to connect the mechanical and electronic parts seems to be rather thick. Usually, there is no appropriate port in the incubators for such a cable. This will make it difficult to put a couple of pumps in an incubator.
About the capability of the pump to provide pulsatile flows more details should be included. What are the needed pulsatile flows in microfluidic applications and what are their technical specifications? Can the pump satisfy those requirements?
The following papers presented similar systems with much more details that everyone can replicate them easily. For instance, the links to SLA files and the used Arduino code are provided. What makes the current work significantly different from them?
1. The FAST Pump, a low-cost, easy to fabricate, SLA-3D-printed peristaltic pump for multi-channel systems in any lab
2. Highly-customizable 3D-printed peristaltic pump kit
Some citations in the introduction seems to be inappropriate:
Line 38: No information about organ-on-a-chip is found in Ref [1].
Line 42: Nothing about the flow patterns and their characteristics in native environments can be found in Ref [4]. Ref [4] is a review article that describes about vascular biology modelling in microfluidics with focus on shear stress in one section. More appropriate reference should be used to clarify the flow patterns or pulsatile nature of flow.
Line 50: Ref [8,9] are review articles that seems to be not much relevant to the mentioned subject. Original articles with more relevant details can be used instead.
Author Response

(The authors gave the same response as above.)

Reviewer 3 Report
Summary: Here the authors developed a 3D printed low-cost peristaltic pump that can be programmed to serve fluid application in microfluidic systems. The authors have quantified the pump performance characteristics of the device. The authors have provided the blue print to build a pump completely and have demonstrated the use case.
Comments: The paper is well written and is a quite timely given the need for fluid pumps for diverse applications in microfluidics. The authors have done a detailed characterization of pump performance and electronic control.
Author Response

(The authors gave the same response as above.)

Round 2
Reviewer 2 Report
The manuscript has been improved by addressing most of the issues appropriately. Well done.